# Chemical Characterization for the Detection of Impurities in Tainted and Natural *Curcuma longa* from India Using LIBS Coupled with PCA

Tejmani Kumar [1], Abhishek Kumar Rai [2], Abhishek Dwivedi [1], Rohit Kumar [3], Mohammad Azam [4], Vinti Singh [5], Neelam Yadav [5] and Awadhesh Kumar Rai [1,*]

1   Laser Spectroscopy Research Laboratory, Department of Physics, University of Allahabad, Prayagraj 211002, India
2   Department of Earth and Planetary Sciences, University of Allahabad, Prayagraj 211002, India
3   Department of Physics, C.M.P. College, University of Allahabad, Prayagraj 211002, India
4   Special Centre for Nanoscience, Jawaharlal Nehru University, New Delhi 110067, India
5   Centre of Food Technology, Institute of Professional Studies, University of Allahabad, Prayagraj 211002, India
*   Correspondence: awadheshkrai@gmail.com

**Abstract:** The present manuscript explores a spectroscopic technique to select turmeric powder, free from impurities, and has compounds of medicinal importance among the tainted and natural turmeric. Six *Curcuma longa* (turmeric powder) samples, named S1, S2, S3, S4, S5, and S6, were analyzed to discriminate between tainted and natural turmeric using the LIBS and multivariate technique. Other techniques such as UV–Vis, FTIR, and EDX are also used to ascertain the elements/compounds showing the medicinal properties of *C. longa*. Spectral lines of carbon, sodium, potassium, magnesium, calcium, iron, strontium, barium, and electronic bands of CN molecules were observed in the LIBS spectra of turmeric samples. Spectral signatures of toxic elements such as lead and chromium are also observed in the LIBS spectra of all samples except S6. Adulteration of metanil yellow, a toxic azo dye, is used to increase the appearance of curcumin when the actual curcumin content is low. The presence of spectral lines of lead and chromium in the LIBS spectra of S1 to S5 suggested that it may be adulterated with lead chromate which is used for coloring turmeric. Further, the presence of sulfur in EDX analysis of sample S5 indicates that it may also have been adulterated with metanil ($C_{18}H_{14}N_3NaO_3S$). The concentration of samples' constituents was evaluated using CF-LIBS, and EDX was used to verify the results obtained by CF-LIBS. The principal component analysis applied to the LIBS data of the turmeric samples has been used for instant discrimination between the sample based on their constituents. We also analyzed antioxidant activity and total phenolic and flavonoid content of different turmeric samples and found a negative Pearson correlation with heavy metals. The presence of curcumin in turmeric is confirmed using LIBS and UV–Vis, which have medicinal properties.

**Keywords:** LIBS; PCA; tainted; natural; *Curcuma longa*





## 1. Introduction

Turmeric (*Curcuma longa*), a member of the Zingiberaceae family, is a spice commonly used to produce traditional dishes in Middle Eastern countries and other parts of Asia. Turmeric is also used as a food preservative, colorant, and dye [1,2]. Spices are well known for their use as flavor enhancement in food, cosmetics, perfume, and serval medicines. Since ancient times, turmeric has been used in medicine because of its potential for antibacterial, anti-inflammatory, antimicrobial, antirheumatic, hypercholesteraemic, antihepatotoxic, antifibroticanti-inflammatory, and insect repellent activity [3–5].

Curcumin is a yellow color compound/molecule present in turmeric ranging from 0.3% to 8.6% [6–9]. It has importance in medicines and cosmetics and can be extracted from

turmeric. It has been reported that factors such as acidity and nutrient content in the soil, cultivation, and type of soil, fertilizer affect the curcumin content in turmeric. Mostly fresh turmeric is free of contamination, but sometimes different types of chemicals are added as substitutes for curcumin [10]. Metanil yellow ($C_{18}H_{14}N_3NaO_3S$), a very toxic azo dye, is mixed with turmeric to imitate the appearance of curcumin. In a joint report by the FAO/WHO committee, metanil yellow comes in the CII category toxicity [11]. However, the long-term toxic effects of metanil are not reported [11].

Recently, the awareness and public concern about the authenticity/adulteration of spices has increased significantly. *C. longa* is the target of adulteration due to its popularity as a component in herbal medicine formulation. Thus, for the sake of quality control of spices/herbal medicines, a rapid and reliable analytical technique must be developed for authentication studies.

Presently several spectroscopic techniques such as atomic absorption spectroscopy (AAS), Raman, FTIR, and UV–Vis spectroscopy are used for quality control [5]. Some other traditional techniques are used to detect the quality of the food which are chromatography-electrospray ionization tandem mass spectrometry, micellar chromatographic method, and high-performance capillary electrophoresis [12–16]. These methods have satisfactory detection limits and high accuracy but still have limited practical application due to their sample destructive nature and operational complexity. Optical methods have been introduced for simple identification quality estimation of foodstuff [13–16]. In the last decade, Laser-induced breakdown spectroscopy (LIBS) emerged as a unique technique that provides accurate, rapid, in-situ, non-destructive, and cost-effective identification of the composition of different materials, including food [17–20]. Multivariate analysis (MVA) is a powerful chemometric analytical tool that takes into account all the possible variables, removes the redundant and correlated variables, and fully utilizes the LIBS spectral information. In chemometrics, many multivariate tools are available for data description, classification, and prediction. The purpose of using MVA is to reduce the dimensions of the spectral data into fewer factors that describe the data. Common chemometric techniques such as principal component analysis (PCA), principal component regression (PCR), partial least square regression (PLSR), and partial least square discriminant analysis (PLS-DA) are applied to identify the distinguishing characteristics of the samples and to build models that describe the relationship between wavelength and their respective intensities in the known samples [21].

Here, LIBS has been used to identify/determine the toxic constituents in the different *C. longa* (turmeric powder) samples harming human health. The present work uses LIBS to identify the essential organic and inorganic elements along with heavy elements Sr and Ba in the turmeric samples, naturally found in plants; but, excess of these elements is also harmful to the human body. A small amount of some toxic elements such as Cr and Pb, in the sample may be very harmful and life-threatening. These have an adverse effect on health and may cause cancer and affect the kidneys, liver, and lungs of the human body. Lead (Pb) increases the high systolic blood pressure and kidney disease, probably carcinogenic to humans, and reduces the IQ level of children. Barium (Ba) may cause a person breathing difficulties. Its high consumption also leads to stomach irritation and swelling of the brain, liver, and heart damage. The regular consumption of metanil yellow present in the sample causes cancer, abnormalities in skin, eyes, lungs, and bone, abnormalities in the fetus, mental retardation, anemia, and accumulation of lead in the body and blood [22]. For better knowledge of the harmful effect of samples on human health, monitoring the exact concentration of elements present in the samples is required. Calibration-free LIBS (CF-LIBS) is suitable to determine the concentration of the constituents of the turmeric sample, as in this technique, the concentration of all the elements can be obtained simultaneously without the use of any reference materials. CF-LIBS results are verified with results obtained from energy-dispersive X-ray spectroscopy (EDX). The organic elements (C, H, O, N) present in the sample help in the identification of toxic compounds/molecules in the sample, such as azo dye metanil yellow and lead chromate.

The FTIR and UV–Vis spectroscopic techniques further confirm the organic molecules present in the sample. EDX data are used to support the quantitative results obtained in the CF-LIBS method. To assist the statements of LIBS and UV–Vis, we have estimated the total phenolic and flavonoid contents and antioxidant activity. Thus, the present manuscript aims to present an analytical tool that is quick, cheap, and nondestructive for rapidly identifying turmeric powder-free from toxic elements/compounds and identifying the compounds responsible for its medicinal properties. One of the multivariate analyses, principal component analysis (PCA), has been used on LIBS data for discrimination of the sample.

## 2. Results

### 2.1. Qualitative Analysis of LIBS Spectra of Curcuma longa

LIBS spectra of all samples S1, S2, S3, S4, S5, and S6 in the wavelength range from 240 to 850 nm were recorded, and typical LIBS spectra of samples S5 and S6 are shown in Figures 1 and 2, respectively. Spectral lines of the inorganic elements (Mn, Ca, Na, K, Mg, Sr, Ba, Al, Pb, Cr) and organic elements (H, C, N, O) are present in the LIBS spectra (Figures 1 and 2) of the turmeric samples. This shows that the samples contain nutritional elements/minerals and organic compounds, along with these the presence of spectral lines of toxic elements are also observed. Here, Cr, and Pb, are toxic to humans if their concentrations are significant. Besides the ionic and atomic spectral lines of the elements, electronic bands of CN, molecules are also observed in the LIBS spectra which further support the presence of organic compounds in the turmeric sample. All the observed atomic and ionic lines of the various elements are tabulated in Table 1. The toxic elements present in the sample may appear due to adulteration or the contaminated soil in which it was grown. Sample S6 does not contain toxic elements such as lead and chromium, the most probable reason for the presence of toxic elements in S1–S5 is adulteration. Based on the intensity of the spectral lines of Pb, Cr, and O in the LIBS spectra and the molecular formula of lead chromate ($PbCrO_4$), this study suggest that S2 contains more lead chromate than other turmeric samples (S1, S3, S4, and S5).

### 2.2. Quantitative Analysis

In the present section, CF-LIBS has been applied to determine the concentration of constituents in the turmeric samples. In the CF-LIBS approach, the concentration of constituents is determined by measuring the intensity of spectral lines of elements present in the sample, which is given by the Boltzmann equation (Equation (1)).

$$I_\lambda^{ki} = C_s\, F\, A_{ki}\, g_k \frac{e^{-(E_k/k_B T)}}{U(T)} \tag{1}$$

where $I_\lambda{}^{ki}$ is the intensity of a spectral line of wavelength $\lambda$, $k_B$ is Boltzmann constant, $C_s$ is the concentration of species, $U(T)$ is partition function, $E_k$ is the upper energy level, and $F$ is an experimental parameter. It is clear from Equation (1) that the concentration of species is directly proportional to the intensity of the spectral line. However, before using the intensity of the spectral line of the elements, which reflects its concentration in the sample, the laser-induced plasma must satisfy three necessary conditions given below.

(i)   Laser ablation should be stoichiometric
(ii)  Laser-induced plasma should be in local thermal equilibrium
(iii) Laser-induced plasma should be optically thin

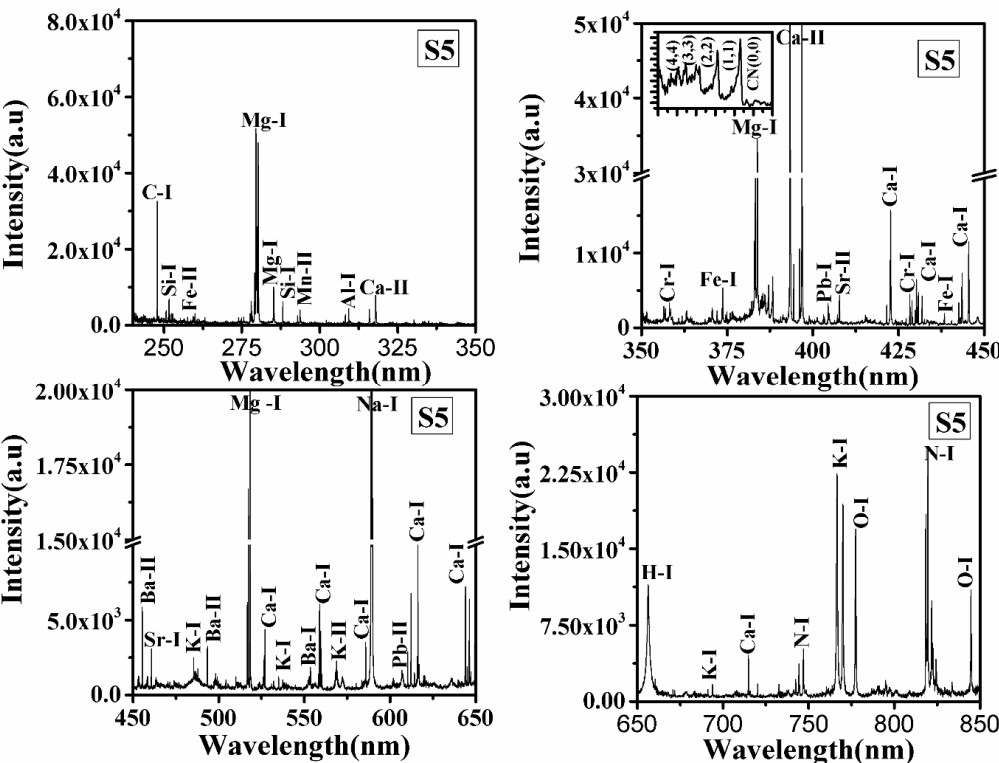

**Figure 1.** LIBS spectrum of turmeric sample (S5) in the spectral range from 240–350 nm, 350–550 nm, 450–650 nm, and 650–850 nm.

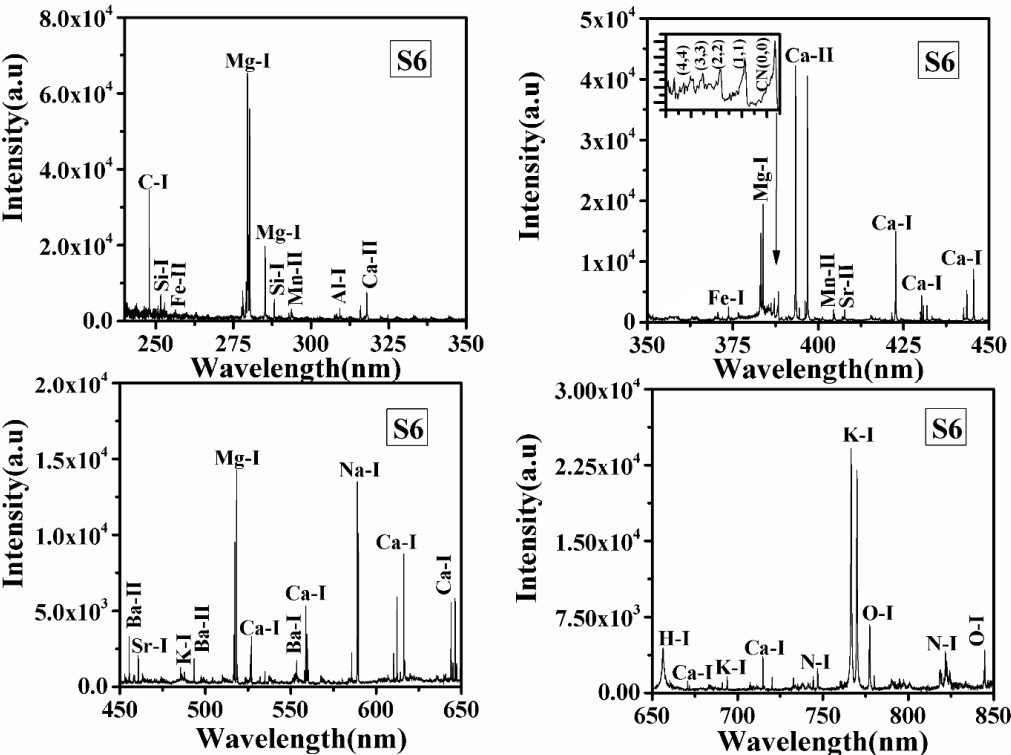

**Figure 2.** LIBS spectrum of natural turmeric sample (S6) in the spectral range from 240–350 nm, 350–550 nm, 450–650 nm, and 650–850 nm.

**Table 1.** Details of the identified elements and molecular band present in samples.

| S. No. | Elements | Wavelength (nm) Observed in LIBS Spectra |
|:---:|:---:|:---:|
| 1. | Carbon (C) | 247.8, |
| 2. | Hydrogen (H) | 656.2 |
| 3. | Oxygen (O) | 777.1, 777.3, 844.5 |
| 4. | Nitrogen (N) | 742.2, 744.1, 746.7, 818.7, 821.5, 824.1 |
| 5. | Calcium (Ca) | 315.8, 317.9, 370.5, 373.6, 393.2, 396.7, 422.6, 430.1, 430.7, 445.4, 442.4, 443.4, 447.9, 485.5, 518.8, 526.1, 526.4, 526.9, 558.1, 559.3, 559.7, 560.0, 560.2, 585.7, 610.1, 612.1, 616.1, 616.8, 643.7, 644.8, 646.1, 647.0, 649.2, 649.8,714.6,720.0,732.4 |
| 6. | Magnesium (Mg) | 277.6, 277.7, 277.9, 278.0, 278.2, 279.0, 279.5, 279.7, 280.2, 285.1, 382.8, 383.1, 383.7, 516.6, 517.2, 518.2, |
| 7. | Sodium (Na) | 588.8, 589.5 |
| 8. | Potassium(K) | 766.3, 769.7 |
| 9. | Strontium (Sr) | 407.7, 421.5, 460.6, |
| 10. | Barium (Ba) | 455.5, 493.4, 553.2, 614.1 |
| 11. | Iron (Fe) | 251.5,252.2, 252.6, 271.4, 271.9, 273.9, 274.9, 275.5, 373.9, 373.7, 374.8, 374.9, 375.8 |
| 12. | Manganese (Mn) | 293.3, 293.9, 294.7,403.1,403.4,404.1 |
| 13. | Aluminium (Al) | 308.1, 309.2, 394.3, 396.1 |
| 14. | Silicon (Si) | 288.1 |
| 15. | Chromium(Cr) * | 357.8, 359.3, 360.5, 425.4, 427.4, 428.9 |
| 16. | Lead(Pb) * | 405.8, 607.5, 608.1 |
| 17. | Molecular bands(CN band) | (0,0),(1,1),(2,2),(3,3),(4,4) |

* lines not observed in S6.

## 2.2.1. Stoichiometric Ablation

The composition of the laser-induced plasma should truly represent the composition of the sample, i.e., the ablation should be stoichiometric. If the laser irradiance is greater than $10^9$ W/cm$^2$ [23], it does not produce much vaporization; instead, an explosion of the small amount (microgram) of the material takes place from the target surface. Since the laser flux density is very high, the ejected material is further heated by absorption of the incoming laser pulse and finally produces plasma. Thus, in this case, the resulting plasma is stoichiometric. The irradiance (power density) at the focused spot of the sample surface is calculated by measuring the diameter of the focused spot using Equation (2) and the irradiance using Equation (3) given below:

$$D = 4\lambda f/\pi d \tag{2}$$

$$\text{Irradiance} = \frac{\text{Power (W)}}{\text{Area (cm}^2)} \tag{3}$$

where $\lambda$ and d are the wavelength and diameter of the laser beam, respectively, and f is the focal length of the lens used to focus the laser beam on the target sample.

The calculated irradiance is $4.07 \times 10^{12}$ W/cm$^2$, which is so much greater than the threshold of $10^9$ W/cm$^2$. Therefore, the laser-induced plasma in the present experiment is stoichiometric.

## 2.2.2. Optically Thin Plasma

In the case of optically thin plasma, self-absorption should not occur, i.e., the emitted photons by the excited atoms are not reabsorbed by similar atoms in the ground state, or the emission from the inner core of the plasma should not be reabsorbed by the outer part of the plasma plume. If the intensity ratio of two interference-free emission lines of the same element having a similar upper state is equal to the product of ratios of the

transition probability, the degeneracy of upper energy level, and the inverse ratio of their wavelengths, then plasma is said to be optically thin [24], i.e.,

$$\frac{I}{I'} = \frac{A_{ki}\, g_k\, \lambda'}{A'_{ki}\, g'_k\, \lambda} \tag{4}$$

where $I$ and $I'$ are the intensities of spectral lines of the same sample element having similar upper states and $A_{ki}$, $g_k$, $\lambda$ and $A'_{ki}$, $g'_k$, $\lambda'$ are corresponding transition probability, the degeneracy of the upper state, and the wavelength of the spectral lines. We have evaluated this for the pair of Ca II at wavelength 393.3 nm/396.8 nm and for Mg II spectral lines at 279.0 nm/279.8 nm. We have estimated the ratio of spectral intensity of these two lines shown in Table 2 and found that the theoretical and experimental values are nearly the same (Table 2).

**Table 2.** Experimental and theoretical intensity of the spectral lines 393.3 nm/396.8 nm for Ca-II and (279.0 nm/279.8 nm) for Mg-II.

| Samples | Intensity Ratio, (I/I') Ca-II (393.3/396.8) | $A_{ki}g_k\lambda'/A'_{ki}g'_k\lambda$ | Intensity Ratio, (I/I') Mg-II (279.0/279.8) | $A_{ki}g_k\lambda'/A'_{ki}g'_k\lambda$ |
|---|---|---|---|---|
| S1 | 1.98 | 1.86 | 0.54 | 0.56 |
| S2 | 2.02 | 1.86 | 0.57 | 0.56 |
| S3 | 1.96 | 1.86 | 0.54 | 0.56 |
| S4 | 2.02 | 1.86 | 0.52 | 0.56 |
| S5 | 1.90 | 1.86 | 0.58 | 0.56 |
| S6 | 1.94 | 1.86 | 0.60 | 0.56 |

### 2.2.3. Local Thermal Equilibrium (LTE)

The plasma is said to be in LTE if it satisfies the necessary and sufficient conditions which are stated below. We have estimated plasma temperature using the following equation.

Taking the logarithm on both sides of Equation (1):

$$\ln\left(\frac{I_\lambda^{Ki}}{g_k A_{ki}}\right) = -\frac{1}{K_B T}E_K + \ln\left(\frac{FC_s}{U(T)}\right) \tag{5}$$

This equation can be written as a linear equation:

$$Y = mX + q_s \tag{6}$$

On comparing both equations; $Y = \ln\left(\frac{I_\lambda^{Ki}}{g_k A_{ki}}\right)$; $X = E_K$; $m = -\frac{1}{K_B T}$ and $q_s = \ln\left(\frac{FC_s}{U(T)}\right)$

The plot of LHS of Equation (6) against $E_K$ for several transitions of any species on the coordinate axis yields a Boltzmann plot (Figure 3) where the slop 'm' of this linear line indirectly gives the plasma temperature.

### 2.2.4. Necessary Condition

If the plasma is in LTE then the electron number density calculated using LIBS spectra must satisfy the McWhirter criterion [25].

McWhirter criterion is based on the value of electron number density present in the laser-induced plasma. The laser-induced plasma should be in LTE if the electron density is greater than the McWhirter limit. According to the McWhirter criterion, the lower limit of electron density is given by the following Equation (7)

$$N_e\ (\text{cm}^{-3}) > 1.6 \times 10^{12}\ [\text{T (K)}]^{1/2}\ [\Delta E(\text{eV})]^3 \tag{7}$$

where $N_e$ is the electron density, T is the temperature in Kelvin, and $\Delta E$ is the maximum energy difference between two states of an atom.

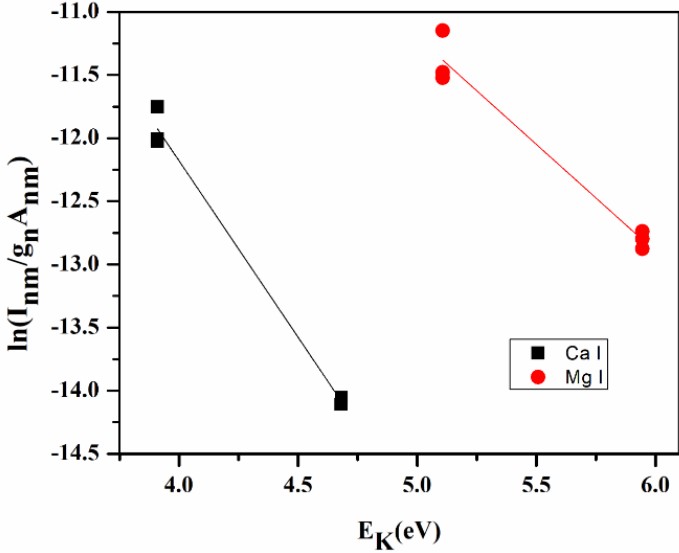

**Figure 3.** Boltzmann plot for turmeric sample (S1).

We can calculate the electron density present in the laser-induced plasma of the sample by measuring the full width at half maxima (FWHM) of spectral lines, as the electron density is related to the FWHM of the Stark broadened line given by Equation (8)

$$N_e \approx 10^{16} \, \Delta\lambda/2w \tag{8}$$

where w is the electron impact parameter obtained from "Principles of Plasma Spectroscopy" by H R Griem [25] and $\Delta\lambda$ is the full width at half maxima (FWHM) of the Stark broadened line (Figure 4).

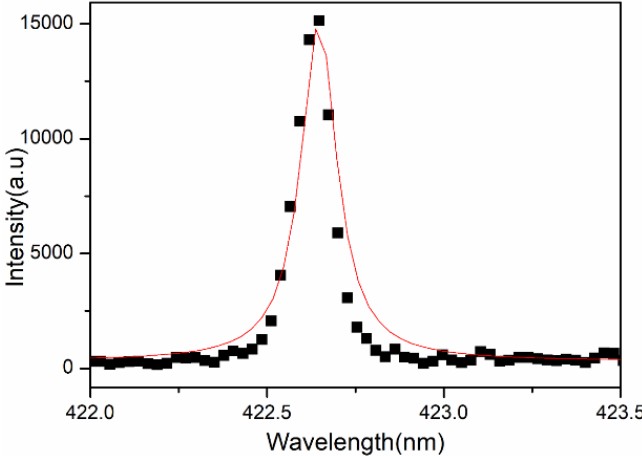

**Figure 4.** Lorentzian plot for estimation of electron density for transmission line.

The plasma temperature from the Boltzmann plot is 10,400 $\pm$ 235 K, and electron density is estimated using Equation (8) is $1.06 \times 10^{18}$ cm$^{-3}$, which is greater than the value of electron density ($1.48 \times 10^{16}$ cm$^{-3}$) for the McWhirter limit (Equation (7)).

It can be observed that the estimated electron density in the LIP of all the samples is greater by two orders of magnitude than the minimum electron density required at any given time, confirming the validity of the LTE criteria over the entire temporal range of the present experiment which fulfills the necessary condition of LTE.

### 2.2.5. Sufficient Condition

If the ionization temperature calculated using the Saha–Boltzmann equation and excitation temperature using the Boltzmann equation coincide within the standard deviation of 15%, the plasma is said to be in LTE.

The Saha–Boltzmann relation of the ionic/atomic emission intensity ratio is given by Equation (9).

$$\ln \frac{I_{ij}^{II} A_{mn}^{I} g_m^{I}}{I_{mn}^{I} A_{ij}^{II} g_i^{II}} = -\frac{\left(E_{ion} - dE + E_i^{II} - E_m^{I}\right)}{kT} + \ln \frac{2(2\pi m_e kT)^{3/2}}{N_e h^3} \tag{9}$$

where $m_e$ is electron mass, $h$ is Planck's constant, $E_{ion}$ is the first ionization potential of the element, $dE$ is the lowering correction parameter, which is the correction term in the first ionization potential arising due to high pressure in the plasma plume, $E_i^{II}$ and $E_m^{I}$ are upper energy levels of ionic and atomic species of the elements having transition probabilities $A_{ij}^{II}$ and $A_{mn}^{I}$ and statistical weights $g_i^{II}$ and $g_m^{I}$.

A graph plotted using Equation (9) is known as the Saha–Boltzmann plot (Figure 5) whose slope is used for the calculation of ionization temperature. If the ionization temperature calculated using the Saha–Boltzmann plot and excitation temperature using the Boltzmann plot coincide within the standard deviation of 15%, the sufficient condition of LTE is satisfied and plasma is said to be in LTE. The estimated temperature calculated by the Saha–Boltzmann plot is 11,340 ± 389 K. This value is very close to the plasma temperature calculated by using the Boltzmann plot with a difference of ~10%.

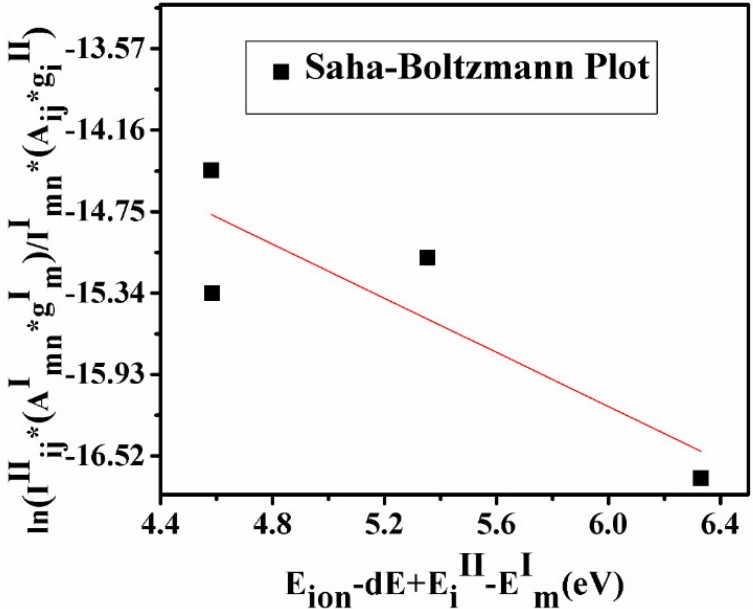

**Figure 5.** Saha–Boltzmann plot to determine the ionization temperature.

After satisfying the above conditions, we can use the spectral intensities of elements present in the samples for the evaluation of their concentration in the sample.

### 2.2.6. Determination of Concentration of Constituents

To compare the concentration of the constituents in turmeric samples, we have measured the spectral lines intensities of various elements. The bar diagram of integrated intensities of the spectral lines of Sr (407.7 nm), Ba (455.4 nm), Cr (357.8 nm), and Pb (607.5 nm) in each sample is shown in Figure 6. The variation in the intensity of the spectral line of Sr and Ba in each sample follows the trends as S5 > S2 > S1 > S4 > S3 > S6. Whereas Cr and Pb follow the trends as S2 > S5 > S4 > S1 > S3. The present experimental results

reveal that the concentration of heavy metals is maximum in S2 and S5 and least in S6. Since Cr, Pb is toxic and harmful, the present work reveals that samples S1, S2, S3, S4, and S5 may be adulterated with lead chromate ($PbCrO_4$). A small concentration of Cr, Pb, is harmful to human health [22]. Therefore, the regular use of sample S2 is very harmful to human health, while S6 is natural and free from toxic elements and is beneficial for human health.

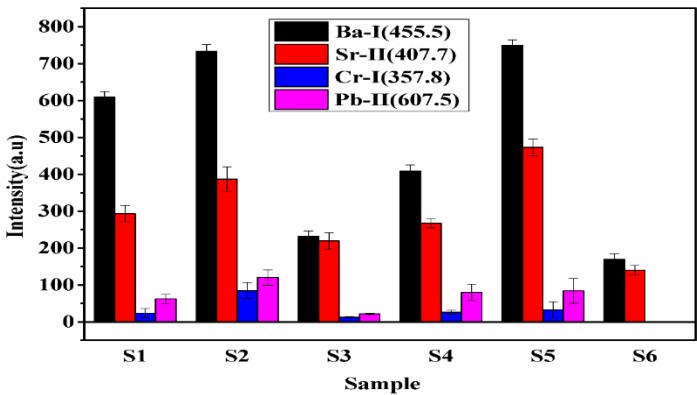

**Figure 6.** The integrated intensities of the spectral line of Ba, Sr, Cr, and Pb elements in the samples.

　　Sample S5 is a local manufacturer of turmeric products, and the market price of this brand is less compared to the other brands. It might be the main motive of this brand is to provide turmeric powder at a low price to beat the other products available in the market. Thus, sample S5 may contain the adulteration of metanil yellow having molecular formula $C_{18}H_{14}N_3NaO_3S$. which is a toxic azo dye to mimic the appearance of curcumin ($C_{21}H_{20}O_6$) when actual curcumin contain is low. The intensity of the spectral lines of Na, C, H, N, and O in sample S5 is higher (Figure 7) as compared to other samples, which reveals that the contents of metanil yellow ($C_{18}H_{14}N_3NaO_3S$) are large in S5. We also see that sample S6 contains more curcumin ($C_{21}H_{20}O_6$) as this turmeric sample is natural as it is collected homegrown without any adulteration. By comparing the spectral intensity of C, N, O, H, Pb, Cr, and Na shown in Figures 6 and 7, we can say that sample S5 is contaminated with metanil azo dye as well as with lead chromate, and sample S2 contain lead chromate whereas sample S6 is pure and natural turmeric powder that contains more curcumin. Curcumin has many scientifically proven health benefits, such as the potential to improve heart health and prevent Alzheimer's and cancer. It is a potent anti-inflammatory and antioxidant and may also help improve symptoms of depression and arthritis. Thus, sample S6 may be used as herbal medicine, in addition to its use as a spice.

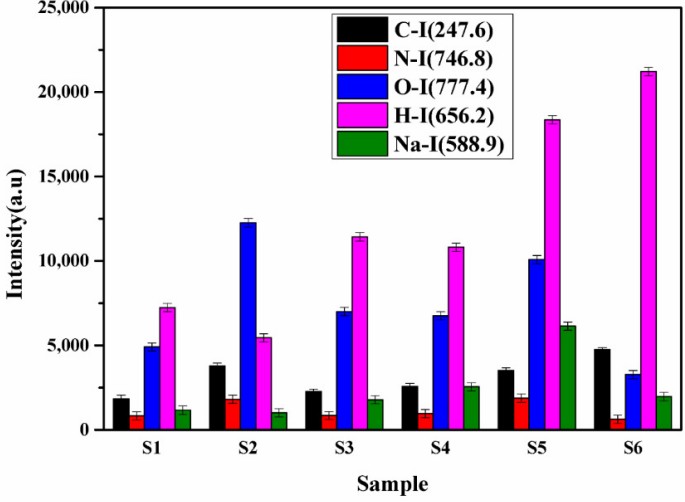

**Figure 7.** The integrated intensities of the spectral line of C, N, O, H, and Na.

Further, the concentration of the constituents is determined using the CF-LIBS approach. Recently calibration-free LIBS (CF-LIBS) has emerged as a satisfactory technique for evaluating the constituents of a variety of materials without using any reference/standard materials/samples [23–25]. Therefore, the adulterated and nutritional elements present in the samples are quantified using the CF-LIBS method. Equation (2) is used to evaluate the concentration of each element present in turmeric samples, and a detailed procedure for the calculation is given in the literature [24]. The concentration of elements present in samples S1–S6 calculated using the CF-LIBS method is summarised in Table 3.

**Table 3.** Concentration of constituents in the turmeric samples using CF-LIBS in weight %.

| Elements | S1 | S2 | S3 | S4 | S5 | S6 |
|---|---|---|---|---|---|---|
| C | 70.9 ± 5.3 | 61.5 ± 4.6 | 66.2 ± 4.7 | 64.1 ± 3.8 | 64.8 ± 4.2 | 72.4 ± 6.4 |
| N | 1.3 ± 0.8 | 2.5 ± 0.9 | 1.4 ± 0.6 | 1.4 ± 0.4 | 2.0 ± 0.3 | 0.9 ± 0.1 |
| O | 12.3 ± 1.3 | 13.8 ± 1.1 | 11.5 ± 1.2 | 10.5 ± 1.3 | 13.8 ± 1.4 | 13.2 ± 1.6 |
| Mg | 0.8 ± 0.02 | 1.3 ± 0.09 | 1.1 ± 0.07 | 0.9 ± 0.02 | 1.2 ± 0.03 | 0.5 ± 0.01 |
| Al | 0.4 ± 0.01 | 0.8 ± 0.02 | 0.6 ± 0.02 | 0.4 ± 0.01 | 0.6 ± 0.01 | 0.8 ± 0.03 |
| Si | 2.1 ± 0.9 | 4.8 ± 0.8 | 9.8 ± 1.2 | 9.3 ± 1.1 | 2.1 ± 0.9 | 6.5 ± 1.3 |
| S | * | * | * | * | * | * |
| K | 2.7 ± 0.8 | 2.4 ± 1.0 | 2.1 ± 0.06 | 3.2 ± 0.9 | 2.3 ± 0.08 | 1.7 ± 0.4 |
| Cr | 0.6 ± 0.02 | 1.1 ± 0.02 | 0.3 ± 0.004 | 0.6 ± 0.007 | 0.7 ± 0.003 | 0 |
| Fe | 0.6 ± 0.02 | 1.3 ± 0.09 | 1.1 ± 0.3 | 0.9 ± 0.04 | 0.5 ± 0.01 | 0.7 ± 0.02 |
| Sr | 1.6 ± 0.3 | 1.9 ± 0.2 | 1.1 ± 0.1 | 1.3 ± 0.4 | 2.1 ± 0.8 | 0.9 ± 0.5 |
| Ba | 1.4 ± 0.2 | 1.7 ± 0.3 | 0.9 ± 0.01 | 0.7 ± 0.02 | 1.6 ± 0.05 | 0.5 ± 0.01 |
| Pb | 3.2 ± 0.1 | 4.6 ± 0.2 | 1.6 ± 0.08 | 3.9 ± 0.09 | 3.8 ± 0.5 | 0 |
| Na | 0.6 ± 0.001 | 0.5 ± 0.002 | 1.1 ± 0.04 | 1.5 ± 0.06 | 2.8 ± 0.09 | 0.9 ± 0.04 |
| Ca | 0.7 ± 0.01 | 1.1 ± 0.04 | 0.8 ± 0.02 | 0.5 ± 0.03 | 1.3 ± 0.05 | 0.7 ± 0.01 |
| Mn | 0.8 ± 0.01 | 0.7 ± 0.02 | 0.4 ± 0.002 | 0.8 ± 0.005 | 0.4 ± 0.002 | 0.5 ± 0.001 |

**\* Indicate element not detected in LIBS spectra**.

### 2.3. Principal Component Analysis (PCA)

PCA was applied to LIBS data to differentiate/distinguish different samples based on their traces and major constituents. The software Unscrambler X is used to discriminate the adulterated and natural/pure *C. longa* (turmeric powder). Sixty LIBS spectra of six *C. longa* (turmeric powder) samples with ten replicas of each are classified into four distinct groups (Figure 8). PC1 separates the samples into two groups. S5 is clustered towards the positive, and S1, S2, S3, S4, and S6 are clustered towards the negative sides of the PC1 axis. On the negative side of PC-1, S2, S4, S1, S6, and S4 form three different clusters too. However, PC2 discriminates the LIBS data with samples S5 and S2, the negative side, and samples S3 and S6 cluster on the positive side of PC2, S1 and S4 lie almost on the PC2 axis as PC2 could not differentiate them properly. Further, the clustering of these samples can be explained by their loading plots of PC1 and PC2 (Figure 8b). In the loading plot of PC1, Ca, Na, and Pb are positively correlated, and Mg is negatively correlated. The clustering of the sample is based on the contents of sodium and magnesium mainly. It is also clear from Figure 7 that the content of sodium is almost double in sample S5 in comparison to others. Thus, S5 is clustered separately from others. PC1 and PC2 explain the total variance of 82% in the data matrix, in which 75% of the variance is along the PC1 axis and 7% of the variance is along the PC2 axis. Thus, the 2D score plot explains that LIBS coupled with PCA may be an important tool for classifying and discriminating *C. longa* (turmeric powder) samples.

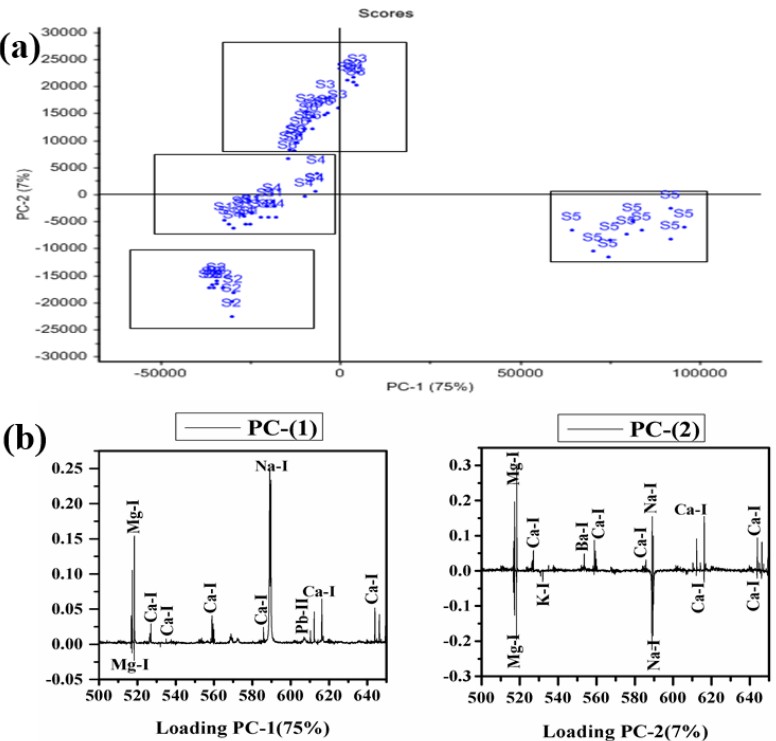

**Figure 8.** PCA score plot along with loading plot of PC1 and PC2 of all samples. (**a**) Score plot and (**b**) Loading plot.

### 2.4. Fourier Transform Infrared Spectra of Turmeric Samples (FTIR)

FTIR analysis is one of the important tools for quickly and efficiently identifying different functional groups present in a sample [5]. FTIR spectra of turmeric samples are recorded in the spectral range 500–3500 cm$^{-1}$ in transmittance mode, as shown in Figure 9. FTIR spectra of all samples contain the absorption bands at 751, 830, 1014, 1160, 1202, 1238, 1286, 1331, 1364, 1419, 1514, 1623, 2950, 2912, and 3341 cm$^{-1}$. A similar type of spectral characteristic of different functional groups has been observed in the spectra of samples S1 to S5. Therefore, our main concern towards the FTIR spectra of samples S5 and S6. Different vibrations associated with different functional groups are given in Table 4. From Figure 9, it has been observed that each vibrational band due to different functional groups is stronger in sample S5 than in sample S6. This may be due to some adulteration in sample S5, which has been taken from the market. Thus, from FTIR spectra, the different vibrational bands are observed due to different groups, which confirm the presence of spectral lines of organic elements C, N, O, and H and the electronic bands of CN molecules as observed in the LIBS spectra.

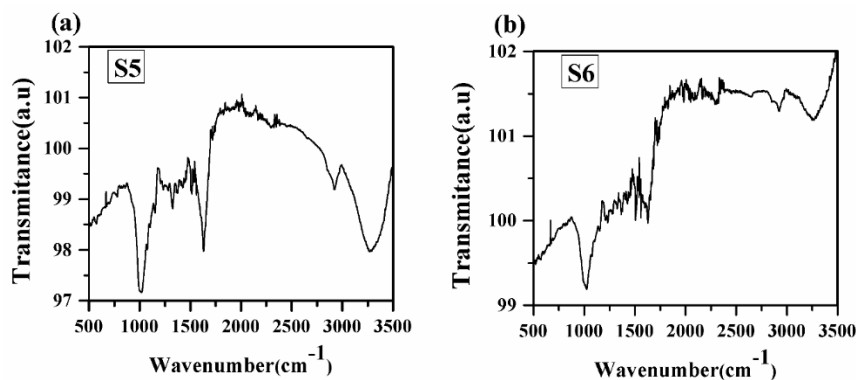

**Figure 9.** FTIR spectrum of turmeric samples S5 and S6. (**a**) Sample (S5); (**b**) Sample (S6).

**Table 4.** Functional group observed in FTIR spectra of turmeric samples.

| S. No. | Wavenumber (cm$^{-1}$) | Functional Group |
|:---:|:---:|:---:|
| 1 | 571 | CH$_2$ stretching |
| 2 | 830 | -HC=CH (Cis) |
| 3 | 1014 | C-OH stretching |
| 4 | 1160, 1202, 1238, 1286 | C-O stretching |
| 5 | 1331, 1364 | CH$_2$ stretching |
| 6 | 1419 | CH$_3$ stretching |
| 7 | 1514 | CN stretching |
| 8 | 1623 | C=O carbonyl group |
| 9 | 2912 | CH$_2$ stretching |
| 10 | 2950 | CH$_3$ stretching |
| 11 | 3341 | OH stretching |

*2.5. UV-VIS Spectroscopy*

It is clear from the LIBS study that the natural turmeric powder S6 does not contain any synthetic color; instead, it contains natural curcumin molecule/compound, which imparts the yellow color of the sample. In contrast, the turmeric powder purchased from the local market contains Pb and Cr which indicate that the manufacturer may have added some synthetic colors such as lead chromate and metanil (azo dye), which impart the brilliant yellow color to these samples. To confirm this statement, we have recorded the UV–Vis spectrum of these samples S1 to S6. Since the absorption spectra of samples S1 to S5 show the same spectral behavior, thus, a typical UV–Vis spectrum of natural turmeric S6 and locally purchased turmeric sample S5 is shown in Figure 10. In Figure 10, the deconvoluted spectra of samples S5 and S6 are shown in which sample S5 shows four different peaks (after deconvolution) instead of two peaks (before deconvolution) and sample S6 shows two peaks (after deconvolution) instead of one peak (before deconvolution). Sample S5 shows peaks at 224, 260, 374, and 425 nm in which peaks at 224 (red peak) and 425 (aqua peak) nm are due to curcumin [26], while peaks at 260 (green peak) and 374 (blue peak) nm are due to lead chromate or metanil [27] as shown in Figure 10a. The UV–Vis peaks due to lead chromate are very strong as compared to the peaks due to curcumin. This suggests the adulteration of lead chromate/metanil in sample S5, which has been purchased from the local market. While in the UV–Vis spectrum of S6 in Figure 10b, two peaks at 224 and 425 nm are observed due to the presence of curcumin, which exists naturally in sample S6 and there is not any type of adulteration. This UV–Vis study verifies the explanation which has been given in the FTIR study about the samples S5 and S6. Thus, the UV–Vis results fairly support the result obtained from the LIBS that the turmeric powders purchased from the market are tainted with harmful chemicals such as lead chromate and metanil to import the brilliant yellow color to attract the consumer.

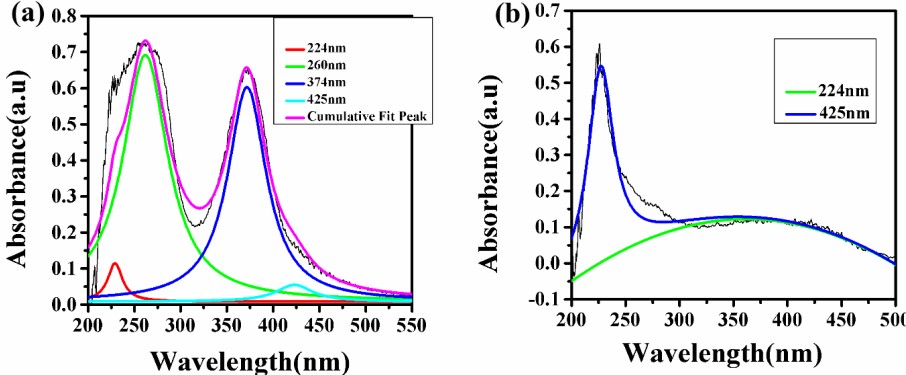

**Figure 10.** UV–Vis spectrum of tainted (S5) and pure (S6) *C. longa* (turmeric powder). (**a**) Sample (S5) (**b**) Sample (S6).

### 2.6. Energy Dispersive X-ray Spectroscopy (EDX)

To confirm the presence of lead chromate and metanil yellow dye in samples (S1–S5) purchased from the local market, we have employed an EDX study. The EDX spectra of S5 and S6 are shown in Figure 11. It is clear from Figure 11a that sample S5 contains the spectral peaks of Pb, Cr, and S (sulfur). These peaks are absent in Figure 11b, corresponding to the EDX spectra of S6. This study also supports the results of LIBS and UV–Vis that the turmeric sample purchased from the local market (S5) is tainted with hazardous chemicals such as lead chromate or metanil yellow.

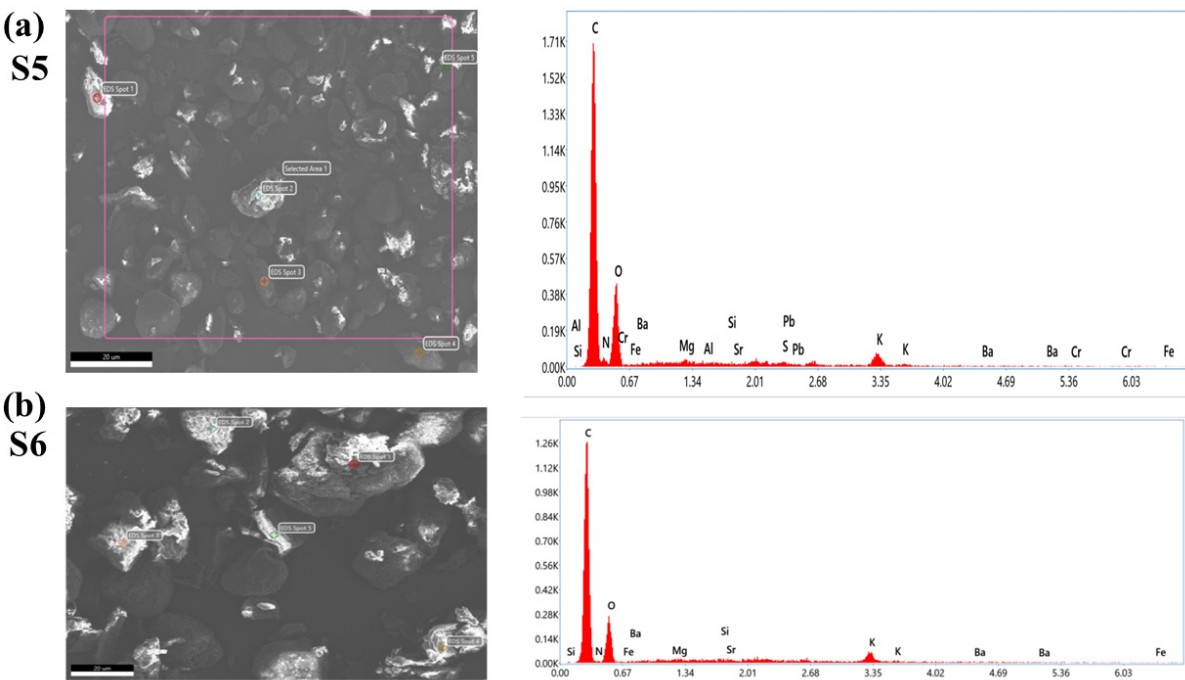

**Figure 11.** EDX spectra of selected turmeric samples (S5) and Sample (S6). (**a**) Sample (S5) (**b**) Sample (S6).

To substantiate the results of the CF-LIBS method, the EDX technique is used to find the concentration of some constituents of the turmeric samples. The same turmeric powder samples were analyzed using an energy-dispersive X-ray spectrometer, and the results are compared in Table 5 and Figure 12. Both the results are in reasonably good agreement.

**Table 5.** The constituents concentrations of elements in *C. longa* (turmeric powder) samples are justified by EDX and CF-LIBS.

| Elements | Sample (S5) EDX Data | Sample (S5) CF-LIBS Data | Sample (S6) EDX Data | Sample (S6) CF-LIBS |
|---|---|---|---|---|
| C | 74.3 ± 0.4 | 64.8 ± 4.2 | 75.8 ± 0.5 | 72.4 ± 6.4 |
| N | 0.8 ± 0.1 | 2 ± 0.3 | 0.6 ± 0.1 | 0.9 ± 0.1 |
| O | 12.3 ± 0.2 | 13.8 ± 1.4 | 15.5 ± 0.2 | 13.2 ± 1.6 |
| Mg | 0.5 ± 0.07 | 1.2 ± 0.03 | 0.1 ± 0.07 | 0.5 ± 0.01 |
| Al | 0.4 ± 0.08 | 0.6 ± 0.07 | 0.6 ± 0.01 | 0.8 ± 0.03 |
| Si | 2.1 ± 0.8 | 2.1 ± 0.9 | 2.4 ± 0.8 | 6.5 ± 1.3 |
| K | 2.7 ± 0.9 | 2.3 ± 0.08 | 3.2 ± 0.9 | 1.7 ± 0.4 |
| Cr | 0.2 ± 0.09 | 0.7 ± 0.003 | 0 | 0 |
| Fe | 0.6 ± 0.09 | 0.5 ± 0.01 | 0.5 ± 0.01 | 0.70 ± 0.02 |
| Sr | 1.2 ± 0.05 | 2.1 ± 0.8 | 0.30 ± 0.09 | 0.9 ± 0.05 |
| Ba | 1.4 ± 0.02 | 1.6 ± 0.05 | 1 ± 0.01 | 0.5 ± 0.01 |
| Pb | 3.2 ± 0.07 | 3.8 ± 0.5 | 0 | 0 |

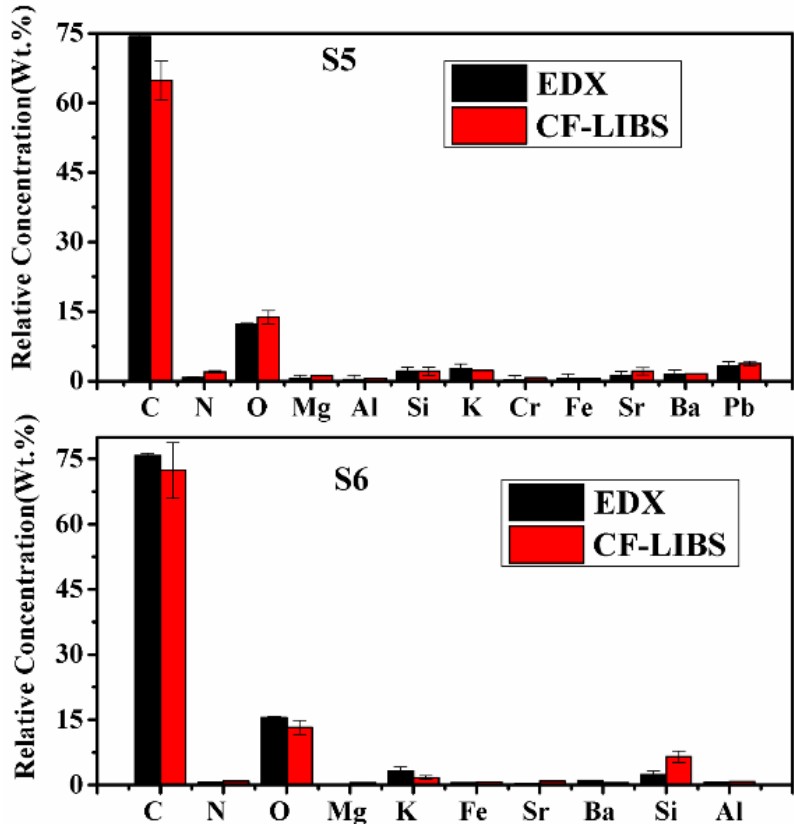

**Figure 12.** Relative concentrations of compositional elements of *C. longa* (turmeric powder) as measured by CF-LIBS and EDX techniques.

The occurrence of heavy elements such as Pb, Cr, Sr, and Ba is also seen in the EDX spectra in Figure 11. These spectra also revealed the presence of major elements as well as trace heavy elements present in the *C. longa* (turmeric powder) sample. These techniques help us to justify the findings of the LIBS analysis.

## 2.7. BioChemical Analysis

The LIBS and UV–Vis spectroscopy technique results demonstrate that the natural turmeric S6 is more effective as an anti-inflammatory and antioxidant than the other turmeric samples S1–S5 as the latter were contaminated with toxic elements/compounds. To validate these statements, we have estimated the total phenolic and flavonoid contents and the estimation of antioxidant activity discussed in the following section.

### 2.7.1. Estimation of Total Phenolic and Flavonoid Contents (TPC and TFC)

Table 6 summarizes the TPC and TFC values of various turmeric extract types. Our result showed that S6 had a 1–2-fold higher concentration of phenolic content (47.61 mg GAE/g extract) than the other turmeric (Table 6). These findings support the LIBS and UV–Vis results that S6 is more suitable than the other samples (S1–S5). In the previous study, Nahak and Sahu [28,29] also compared the phenolic content of several turmeric species and revealed that *C. longa* had the maximum TPC content, followed by other varieties of turmeric. In 2014, Denre reported the TPC value of 38 mg GAE/g dry turmeric of *C. longa* from West Bengal [28] which was greater than S1 to S5 turmeric samples but lower than S6. Like TPC, S6, had significantly greater flavonoids than the other samples (Table 6), which is partially compatible with Alafiatayo et al. and Tanvir et al. [30,31].

**Table 6.** Total phenolic content (TPC), total flavonoid contents (TFC), FRAP, and DPPH values measured in different varieties of turmeric samples.

| S. No. | Sample Code | TPC (mg GAE/g) | TFC (mg QE/g) | DPPH (%) | FRAP (mg QE/g) |
|---|---|---|---|---|---|
| 1. | S1 | 31.91 ± 0.55 [b] | 65.43 ± 0.73 [b] | 60.41 ± 1.02 [c] | 50.8 ± 1.46 [e] |
| 2. | S2 | 20.20 ± 1.00 [a] | 54.43 ± 0.70 [a] | 54.17 ± 0.63 [a] | 25.4 ± 0.72 [b] |
| 3. | S3 | 38.20 ± 0.64 [c] | 76.01 ± 0.58 [c] | 60.07 ± 1.00 [c] | 37.2 ± 0.64 [c] |
| 4. | S4 | 32.28 ± 0.51 [b] | 76.84 ± 0.92 [c] | 58.36 ± 0.70 [b] | 44.3 ± 0.68 [d] |
| 5. | S5 | 31.52 ± 0.50 [b] | 95.81 ± 0.90 [d] | 57.28 ± 0.63 [b] | 23.5 ± 0.74 [a] |
| 6. | S6 | 47.61 ± 0.81 [d] | 116.21 ± 0.64 [e] | 70.68 ± 0.84 [d] | 55.8 ± 0.92 [f] |

Data are expressed as the mean ± standard error of mean (SEM) of three independent experiments in triplicate. Different letters (a, b, c, d, e, and f) in each column indicate a significant difference ($p < 0.05$) according to Duncan's multiple range tests.

2.7.2. Estimation of Antioxidant Activity

In our study, S6 showed the highest scavenging activity (70.68 ± 0.84%), which was significantly different from other samples (Table 6). Our findings are consistent with the observations of the LIBS and UV–Vis and with the work of Alafiatayo et al. [30] who studied the antioxidant potential of several turmeric varieties. These findings are also in agreement with those of Nahak and Sahu [29], who evaluated the antioxidant activity of ethanolic extracts of different turmeric samples and found that *C. longa* had the highest antioxidant activity. Among the six turmerics, the highest reducing power ferric reducing antioxidant power (FRAP) was observed in S6 followed by S3 and S1. Similar to diphenylpicrylhdrazyl (DPPH), S6 observed a significantly higher FRAP value (55.8 ± 0.92 mg QE/g), whereas S2 had the lowest value (25.4 ± 0.72 mg QE/g extract). The FRAP value of *C. longa* from Thailand was 815.4 μmol TE/g dry weight [32], which was much lower than S6 but higher than S2. The phenolic content and antioxidant activity of different species and cultivars of turmeric (Curcuma spp.) were also reported by Akter et al. [32] with results that are remarkably similar to ours.

Biochemical analysis showed that the S6 sample contained the highest amount of total phenolic content (TPC) and total flavonoid content (TFC) among turmeric samples; the reason behind this can be as it was a fresh raw turmeric sample, which was processed in lab conditions, while other samples were purchased from the market, and mineral analysis results indicate the presence of some adulterants, which may cause changes in secondary metabolites. Table 7 presents the correlation matrix (Pearson correlation coefficients) among the selected heavy metals concentration, TPC, TFC, and antioxidant activity. The significantly higher antioxidant activity of S6 may be attributed to the higher TPC and TFC as we found a very strong positive correlation between TPC (R = 0.928) and TFC (R = 0.789) of turmeric with their antioxidant activity. Kim et al. [33,34] also observed high correlations between the TPC and TFC, and DPPH radical scavenging activity in various spices, including turmeric. Meanwhile, all heavy metals showed a negative correlation with TPC, TFC, DPPH, and FRAP radical scavenging activity. B. Marquez-Garc, et al. [35] revealed a similar pattern in *Erica andevalensis*. They explained it with two reasons: the first is that certain levels of heavy metals may increase the phenolic compound concentration because phenoxyl radicals resulting from antioxidative reactions could act as prooxidants [36]. However, when plant materials are exposed to the highest concentration of metals may reduce the synthesis or release of phenolics by an unknown mechanism to avoid a deleterious effect caused by the phenoxyl radicals produced. According to the second hypothesis, an overabundance of metals may have hindered the antioxidative system responses based on phenolics and other substances, resulting in the inability to produce new phenols. A. Perna et al. [37] also reported the same trend in different honey samples. They found correlation coefficients between total flavonoid and metal content were low and not significant.

**Table 7.** Correlation matrix (Pearson correlation coefficients) among the considered parameters.

|      | TPC | TFC | DPPH | FRAP | SrII | CrI | PbII |
|------|-----|-----|------|------|------|-----|------|
| **TPC**  | 1 | | | | | | |
| **TFC**  | 0.830 * | 1 | | | | | |
| **DPPH** | 0.928 ** | 0.789 | 1 | | | | |
| **FRAP** | 0.695 | 0.360 | 0.796 | 1 | | | |
| **SrII** | −0.763 | −0.360 | −0.789 | −0.847 * | 1 | | |
| **CrI**  | −0.929 ** | −0.700 | −0.782 | −0.697 | 0.667 | 1 | |
| **PbII** | −0.964 ** | −0.680 | −0.885 * | −0.689 | 0.819 * | 0.897 * | 1 |

\* Correlation is significant at the 0.05 level (2-tailed). \*\* Correlation is significant at the 0.01 level (2-tailed).

## 3. Material and Methods

### 3.1. Sample Collection and Preparation for LIBS

Five commercially available turmeric powder samples (named S1, S2, S3, S4, and S5) were collected randomly from the spice market of Allahabad, India. The sixth sample (natural/unprocessed, named S6) was procured from the farmer. To record the LIBS spectra, 0.60 g of each sample was taken, and the pellet was prepared with the help of a hydraulic pressure machine (K-Br Press MODEL M-15). Pellets of different samples, shown in Figure 13, were directly used to record the LIBS spectra of the samples using the experimental arrangement shown in Figure 14.

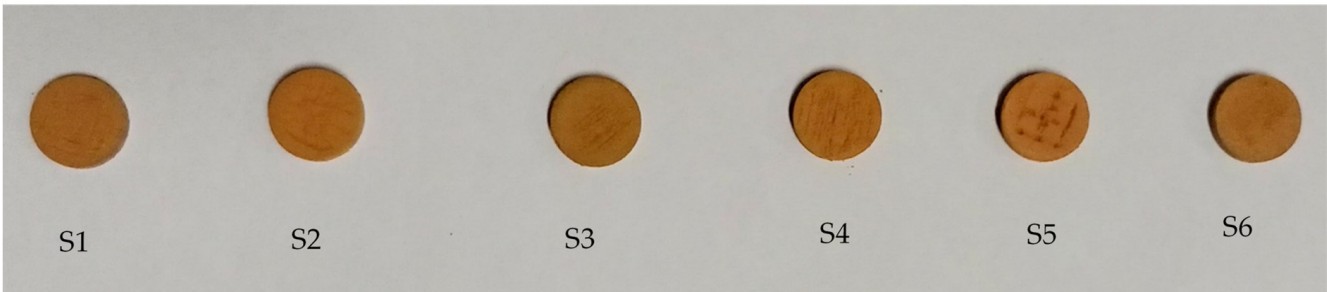

**Figure 13.** Pellet of different turmeric samples.

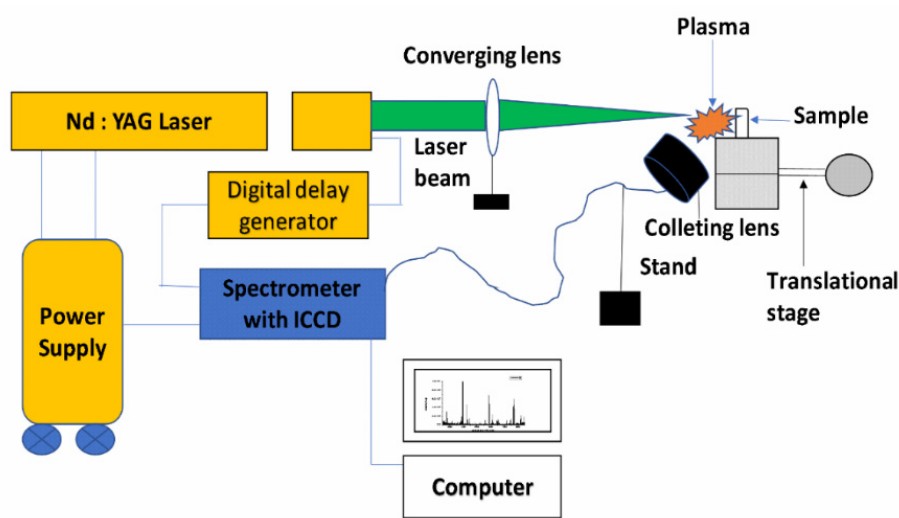

**Figure 14.** Schematic diagram of the experimental setup for the LIBS technique.

High power pulsed Nd: YAG laser (Continuum Surelite III-10), having a variable repetition rate from 1Hz to 10 Hz with pulse duration of 4 ns and maximum energy 425 mJ at 532 nm wavelength, was focused on the sample using a plano-convex quartz lens of 15 cm focal length to produce the laser-induced plasma on the sample surface. The pellets

of each turmeric powder sample were placed at the movable translation stage, which was capable of moving in three X, Y, and Z directions to get fresh laser shots so that there was no chance of crater formation on pellets.

The emission from the plasma was collected at a≈45° angle to the incident laser beam by using an optical fiber system and fed into the Mechelle spectrometer (Mechelle ME 5000 Andor) equipped with an intensified charge-coupled device (ICCD istar 734 Andor). The 11 mJ per pulse of laser energy, repetition rate of 2 Hz, gate delay of 1 μs, and gate width of 5 μs were found to be suitable at which the best signal-to-noise and signal-to-background ratios were observed in the LIBS spectra. Accumulation of 50 laser pulses was used to produce one LIB spectrum. Ten such spectra of each sample were recorded and averaged for analysis using Andor SOLIS Software. Spectral lines of various elements were identified with the help of the National Institute of Standard and Technology (NIST) Atomic Spectroscopy Database (ASD) [38], and for molecular band identification, "The identification of Molecular Spectra" was used [39].

### 3.2. Experimental Setup for UV-VIS

The UV–Vis spectra of the turmeric samples were recorded in the range from 200 to 900 nm using a UV–Vis spectrophotometer (model Systronics–2202). The same sample of turmeric powder was used in solution form using distilled water as a solvent.

### 3.3. Experimental Set-Up for EDX

EDX spectra of turmeric samples were recorded using Field emission scanning electron microscope-FESEM (Model FEI NOVA NANO SEM-450). The same powder form of the sample was used for EDX analysis.

### 3.4. Experimental Set-Up for FTIR

FT–IR spectra of turmeric samples were recorded in the range from 400 to 4000 cm$^{-1}$ using PerkinElmer FT–IR/FIR Frontier spectrometer. The powder form of the same sample was used for recording the FTIR spectra in attenuated total reflectance (ATR) mode.

### 3.5. Biochemical Analysis

3.5.1. Preparation of Extract

An identical amount of different turmeric powder (20 g) was extracted with methanol (200 mL) for 48 h at a temperature of 25 °C with constant magnetic stirring. The solutions were then filtered through Whatman No. 1 two-fold filter paper. The process was repeated three times with fresh MeOH added to the used plant material. The filtered solutions were dried at 40 °C in a rotary evaporator under decreased pressure. All extracts were maintained at 4 °C in the refrigerator.

3.5.2. Determination of Total Phenolic Content, Flavonoid Contents, and Antioxidant Activity

The Folin–Ciocalteu method, as modified by Beretta et al. [40], was used to determine the total phenolic content. This content was expressed as mg of gallic acid equivalent (GAE) per g sample. Total flavonoid contents (TFC) was determined using the method described by Djeridane et al., 2006 [41], and the results were expressed as mg of quercetin equivalent (QE) per g test sample.

Antioxidant activity was analyzed with DPPH radicals and ferric reducing antioxidant power (FRAP) assay. When the DPPH radical (DPPH•) reacts with an antioxidant, it is reduced. Using a UV–Vis spectrophotometer, the color shifts (from deep violet to light yellow) were measured at 517 nm. The DPPH radical scavenging activity of the samples was measured using a protocol defined by Beretta et al. [40] with certain changes. The FRAP assay detects the change in absorbance at 593 nm caused by electron-donating antioxidants forming blue-colored $Fe^{2+}$ TPTZ from colorless oxidized $Fe^{3+}$ TPTZ. The assay was carried out using Bertoncelj et al.'s [42] methodology, with some changes. For the calibration curve,

aqueous standard solutions of $FeSO_4$ $7H_2O$ (100–1000 µm) were adopted, and the results were represented as the FRAP value (µm Fe(II)) of the solution.

## 4. Discussion and Conclusions

The qualitative and quantitative analyses of all turmeric samples have been examined using LIBS, UV–Vis, FTIR, EDX, and biochemical analysis techniques. The results of the LIBS analysis reveal the presence of inorganic and organic elements in the samples of turmeric, revealing that turmeric contains important minerals and some organic compounds which ultimately have medicinal properties. LIBS results also show the presence of heavy metals in all turmeric samples purchased from the local market, which are harmful to humans consuming them. Further, the results show that sample S6, natural turmeric powder, is minimal/negligible toxic in the presence of heavy metals. The presence of the spectral lines of organic elements in S6 is probably the presence of curcumin which is responsible for the medicinal properties of turmeric. The correlation between the molecular formula of the metanil and the spectral lines of the elements present in the LIBS spectra and EDX of turmeric powder (S5) confirms the presence of metanil azo dye compounds in the turmeric sample (S5). Chromium and lead contents are more in sample S2 as compared to other samples, which reveals that it (sample S2) may be more adulterated with lead chromate. The presence/composition of the organic compounds/molecules in different turmeric samples are further confirmed by using UV–Vis and FTIR techniques. EDX spectroscopy is also used to confirm the existence of toxic metals in the turmeric samples. The concentration of all constituents is determined using the CF-LIBS technique, which is validated by the results obtained by EDX. Both the results are in good agreement. All the turmeric samples have been discriminated using PCA, which distinguishes the turmeric sample into four groups samples S1 and S4 cluster in one place, similarly samples S3 and S6 in another place, and S2 and S5 are clustered in different places. The absorption bands of curcumin ($C_{12}H_2O_6$) present in UV–Vis are also confirmed by the functional group of $CH_2$, $CH_3$, C=O, and CN stretching in FTIR spectra which show the presence of curcumin molecules in the sample. Although the absorption bands of phenolics and flavonoids are not observed in the UV–Vis spectra, the stretching of the hydroxyl group (O-H) and H-bonded stretching, which is characteristic of polyphenolic compounds and C-O stretching due to the typical flavonoids contents are observed in the FTIR spectra of the sample. The above observation may indicate that the phenolics compound and flavonoids are present in the sample. Spectral lines of the inorganic elements (Mn, Ca, Na, Fe, Si, K, Mg, Sr, Ba, Al, Pb, Cr) and organic elements (H, C, N, O) are present in the LIBS spectra has been also confirmed by the EDX methods which also reveals the presence of organic elements (C, N, O) and inorganic elements Mg, Si, K, Cr, Fe, Sr, Br, Pb, Al (including sulphur (S) which are not seen in LIBS.

The experimental results of the study reveal that turmeric samples show high levels of phenolics and flavonoids, which function as effective natural antioxidants. The relationships between metal presence and total phenolic and antioxidant activities were especially compelling, revealing that minerals may have an impact on polyphenol levels in turmeric samples. Thus, the experimental results of the present work also indicate that S1 to S5 may contain lead chromate. Additionally, S5 contains metanil azo dye which is very harmful to human health and sample S6 is free from these toxic elements and compounds which is more suitable for human health and wellness.

**Author Contributions:** Conceptualization, T.K.; data curation, T.K.; formal analysis, T.K., R.K. and M.A.; investigation, T.K.; resources, T.K., A.K.R. (Abhishek Kumar Rai) and A.K.R. (Awadhesh Kumar Rai); supervision, A.K.R. (Awadhesh Kumar Rai); validation, R.K. and M.A.; visualization, T.K.; writing—original draft, T.K.; writing—review and editing, A.K.R. (Abhishek Kumar Rai), A.D., V.S., N.Y. and A.K.R. (Awadhesh Kumar Rai). All authors have read and agreed to the published version of the manuscript.

**Funding:** This research received no external funding.

**Institutional Review Board Statement:** Not applicable.

**Informed Consent Statement:** Not applicable.

**Data Availability Statement:** Not applicable.

**Acknowledgments:** The authors thank P. K. Shahi for his fruitful suggestions during the preparation of the manuscript. Two authors T.K. and A.D. thank UGC, New Delhi for CRET and D. S. Kothari fellowships as financial support.

**Conflicts of Interest:** The authors declare no conflict of interest.

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
