# Peer review of "Chemical Characterization for the Detection of Impurities in Tainted and Natural Curcuma longa from India Using LIBS Coupled with PCA"

_atoms, doi:10.3390/atoms10030091_

Round 1

Reviewer 1 Report

Overall, this paper is quite interesting, analyzing commercial turmeric samples sole in India using various available analytical techniques with a special emphasis on the use of laser induced breakdown spectroscopy. However, I found several things that need to be improved before this manuscript can be accepted for publication. The first point is about writing. There are a lot of abbreviations found in the text that appear without being preceded by an extension. I think all these abbreviations should be written down the first time they appear in writing, and then be abbreviated on subsequent occasions. I also found the use of capitals, conjunctions and punctuation that was not very appropriate on several occasions in the manuscript draft.

On pages 4 dan 5, the y-axix scale of Figures 1 and 2 should be made the same. 

The second point is regarding the plasma properties, although the importance and how the temperature measurement carried out was described, pp. 8-9, the plasma temperature has not been disclosed in this manuscript. Likewise, the electron density of plasma, has not been disclosed in this manuscript. Similarly, Saha-Boltzmann plot, mentioned in the manuscript, p. 10, but no such plot and estimated temperature found in the manuscript. I think these need to be written clearly in the manuscript so that readers can better evaluate the condition of the plasma. For the results shown in Table 3, what are the units? PCA results need to be discussed more carefully, especially why S1, S2, S3, S4 and S6 are in one cluster, even though the LIBS measurement results show that only 1 sample contains Pb, namely S6. I think this needs to be considered carefully. In Figure 7 it is necessary to give a clear indication, which is 7(a) and which is 7(b).

In the discussion of section 2.4., p. 13, the figure number is wrong, it was written "Figure 10", while the correct figure number is Figure 8. Figure 8, FTIR spectrum, also needs to be made a clear indication, which one is 8(a) and which one is 8(b). The scale of the axes in the two figures also needs to be the same. Likewise for Figure 9, it is necessary to have a clear indication, which are (a) and (b).

page 16 in the first paragraph, it is not clear which picture is meant, it needs to be clarified page 18, the unit of mass used should be the international system, namely "g" not "gm". page 20, I think the research results are much interesting if we can try to draw a clearer thread between the results of elemental analysis with LIBS confirmed by EDX and the results of molecular analysis using UV Vis and FTIR, especially with regard to the content of curcuminpage 16 in the first paragraph, it is not clear which picture is meant, it needs to be clarified

page 18, the unit of mass used should be the international system, namely "g" not "gm". for the discussion section on page 20, I think the research results will be much more interesting and meaningful if it can be tried to systematically explain the relationship obtained between the results of elemental analysis with LIBS confirmed by EDX and the results of molecular analysis using UV Vis and FTIR, especially with regard to the content of curcumin or even with the content of phenolics and flavonoids

Author Response

      Response to Reviewer 1 Comments

Point 1: Overall, this paper is quite interesting, analyzing commercial turmeric samples sole in India using various available analytical techniques with a special emphasis on the use of laser induced breakdown spectroscopy. However, I found several things that need to be improved before this manuscript can be accepted for publication. The first point is about writing. There are a lot of abbreviations found in the text that appear without being preceded by an extension. I think all these abbreviations should be written down the first time they appear in writing, and then be abbreviated on subsequent occasions. I also found the use of capitals, conjunctions and punctuation that was not very appropriate on several occasions in the manuscript draft.

Response 1: We thank the learned reviewer for the suggestions. We have incorporated changes according to the suggestions in the revised manuscript.

Point 2: On pages 4 dan 5, the y-axis scale of Figures 1 and 2 should be made the same.

Response 2: In the original manuscript,  we have chosen the scale on the Y-axis so  that weak spectral lines of the elements could also be visible in Figures 1 and 2. But considering the learned reviewer’s suggestions, we have made the Y-axis of figure 1 the same as figure 2.

Point 3: The second point is regarding the plasma properties, although the importance and how the temperature measurement carried out was described, pp. 8-9, the plasma temperature has not been disclosed in this manuscript. Likewise, the electron density of plasma, has not been disclosed in this manuscript. Similarly, Saha-Boltzmann plot, mentioned in the manuscript, p. 10, but no such plot and estimated temperature found in the manuscript. I think these need to be written clearly in the manuscript so that readers can better evaluate the condition of the plasma. 

Response 3: We have incorporated changes according to the suggestions.

The plasma temperature is 10400±235K, and electron density is estimated as 1.06X1018cm-3which is greater than the value of electron density (1.48X1016cm-3) of the McWhirter limit. The Saha Boltzmann plot is incorporated in the manuscripts, and the estimated temperature calculated by the Saha-Boltzmann plot is 11340±389K. This value is very close to the plasma temperature calculated by using the Boltzmann plot with a difference of ~10%.

All the above changes have been incorporated in the revised manuscripts.

Point 4: For the results shown in Table 3, what are the units?

Response 4: The results shown in table 3 are in Weight%.

Point 5: PCA results need to be discussed more carefully, especially why S1, S2, S3, S4 and S6 are in one cluster, even though the LIBS measurement results show that only 1 sample contains Pb, namely S6.I think this needs to be considered carefully.

Response 5: The sample S6 is not separately clustered from the others sample due to the effect of the other matrix elements such (Ca, Na, Mg…etc). This is more clear from the loading plot shown in figure 8(b); the clustering of the sample is based on contents of sodium and magnesium. It is clear from figure 7 (bar plot) that the content of sodium is almost double in sample S5 in comparison to others .it is also clear from the score plot Figure 8 (a) S5 forms a separate cluster.

Point 6: In Figure 7 it is necessary to give a clear indication, which is 7(a) and which is 7(b).

Response 6: We have modified the figure accordingly.

Point 7: In the discussion of section 2.4., p. 13, the figure number is wrong, it was written "Figure 10", while the correct figure number is Figure 8. Figure 8, FTIR spectrum, also needs to be made a clear indication, which one is 8(a) and which one is 8(b). The scale of the axes in the two figures also needs to be the same. Likewise for Figure 9, it is necessary to have a clear indication, which are (a) and (b).

Response 7: As per reviewer suggestions, we have modified the figure and text.

Point 8: page 16 in the first paragraph, it is not clear which picture is meant,

Response 8: We have modified the text.

 Point 9:it needs to be clarified page 18, the unit of mass used should be the international system, namely "g" not "gm".

Response 9: We have modified the unit of mass as ‘g’ instead of ‘gm’.

Point 10: page 20, I think the research results are much interesting if we can try to draw a clearer thread between the results of elemental analysis with LIBS confirmed by EDX and the results of molecular analysis using UV Vis and FTIR, especially with regard to the content of curcumin or even with the content ofphenolics and flavonoids

Response 10: We agree with the learned reviewer’s comment and incorporated changes according to the suggestions.

The absorption bands of curcumin (C12H2O6) present in UV-Visible are also confirmed by the functional group of CH2, CH3, C=O, and CN stretching in FTIR spectra which show the presence of curcumin molecules in the sample.

Although the absorption bands of phenolics and flavonoids are not observed in the UV-Visible spectra but the stretching of hydroxyl group(O-H) and H-bonded stretching which is characteristic of polyphenolic compounds and C-O stretching due to the typical flavonoids contents are observed in the FTIR spectra of the sample. The above observation may indicate that the phenolics compound and flavonoids are present in the sample

Spectral lines of the inorganic elements (Mn, Ca, Na, Fe, Si, K, Mg, Sr, Ba, Al, Pb, Cr) and organic elements (H, C, N, O) are present in the LIBS spectra has been also confirmed by the EDX methods which also reveals the presence of organic elements (C, N, O) and inorganic elements Mg, Si, K, Cr, Fe, Sr, Br, Pb, Al (including sulphur(S) which are not seen in LIBS.                

Point 11: for the discussion section on page 20, I think the research results will be much more interesting and meaningful if it can be tried to systematically explain the relationship obtained between the results of elemental analysis with LIBS confirmed by EDX and the results of molecular analysis using UV Vis and FTIR, especially with regard to the content of curcumin or even with the content of phenolics and flavonoids

Response 11:  We have incorporated changes according to the suggestions.

Reviewer 2 Report

The authors discussed here the '' Exploration and detection of impurities and compounds showing medicinal properties in tainted and natural Curcuma longa using different analytical techniques'', where the authors analysis the chemical and toxicity of the tainted and natural turmeric powder commercial available in India using PCA on LIBS technigues.  The results present different elements that are found in the poweders and these where detected using mainly PCA-LIBS rather than the conventional spectroscopic  such as AAS. This makes this paper unique. First of all I think that this paper is, as a whole, scientifically sound and the topic is of wide interest especial for the scientist who are interested in using PCA on LIBS.  However, in my opinion, there are points that need to be rectified prior to publication:

Title: There should be only one meaning to the title. It should accurately describe the study. The words "medical properties" in the title is meaningless. The article is about the chemical characterization of the tainted and natural turmeric from India. There must be only one meaning to the title. The topic (chemical analysis of turmeric), the protocol (PCA-LIBS), a summary of the results (tainted turmeric is contaminated compare to natural one), and the population studied (Turmeric from India) must be mentioned in the title. So the title must be revised.

Most of the abbreviations in the text are not defined such as PCA, PCR, PLSR, and PLS‐DA.

Thanks

Author Response

    Response to Reviewer 2 Comments

The authors discussed here the '' Exploration and detection of impurities and compounds showing medicinal properties in tainted and natural Curcuma longa using different analytical techniques'', where the authors analysis the chemical and toxicity of the tainted and natural turmeric powder commercial available in India using PCA on LIBS technigues.  The results present different elements that are found in the poweders and these where detected using mainly PCA-LIBS rather than the conventional spectroscopic such as AAS. This makes this paper unique. First of all I think that this paper is, as a whole, scientifically sound and the topic is of wide interest especial for the scientist who are interested in using PCA on LIBS.  However, in my opinion, there are points that need to be rectified prior to publication:

Point 1:   Title: There should be only one meaning to the title. It should accurately describe the study. The words "medical properties" in the title is meaningless. The article is about the chemical characterization of the tainted and natural turmeric from India. There must be only one meaning to the title. The topic (chemical analysis of turmeric), the protocol (PCA-LIBS), a summary of the results (tainted turmeric is contaminated compare to natural one), and the population studied (Turmeric from India) must be mentioned in the title. So the title must be revised.

Response 1: We have incorporated changes according to the suggestions. The title of the manuscript is revised as “Chemical characterization for the detection of impurities in tainted and natural Curcuma longa from India using LIBS coupled with PCA’’.

Point 2: Most of the abbreviations in the text are not defined such as PCA, PCR, PLSR, and PLS‐DA.

 Response 2: We agree with the learned reviewer’s comment and incorporated changes according to the suggestions.

PCA-Principal Component Analysis

PCR-Principal Component Regression

PLSR-Partial Least Square Regression

PLS-DA: Partial Least Squares Discriminant Analysis

The above modification is also included in the modified manuscript.
